# Investigating Intersectional Bias in Large Language Models using Confidence Disparities in Coreference Resolution

**Falaah Arif Khan**[1], **Nivedha Sivakumar**[2], **Yinong Oliver Wang**[1], **Katherine Metcalf**[2],
**Cezanne Camacho**[2], **Barry-John Theobald**[2], **Luca Zappella**[2], **Nicholas Apostoloff** [2],
[1]Work done while at Apple, [2]Apple,

## Abstract

Large language models (LLMs) have achieved impressive performance, leading to their widespread adoption as decision-support tools in resource-constrained contexts like hiring and admissions. There is, however, scientific consensus that AI systems can reflect and exacerbate societal biases, raising concerns about identity-based harm when used in critical social contexts. Prior work has laid a solid foundation for assessing bias in LLMs by evaluating demographic disparities in different language reasoning tasks. In this work, we extend single-axis fairness evaluations to examine intersectional bias, recognizing that when multiple axes of discrimination intersect, they create distinct patterns of disadvantage. We create a new benchmark called WinoIdentity by augmenting the WinoBias dataset with 25 demographic markers across 10 attributes, including age, nationality, and race, intersected with binary gender, yielding 245,700 prompts to evaluate 50 distinct bias patterns. Focusing on harms of omission due to underrepresentation, we investigate bias through the lens of uncertainty and propose a group (un)fairness metric called *Coreference Confidence Disparity* which measures whether models are more or less confident for some intersectional identities than others. We evaluate five recently published LLMs and find confidence disparities as high as 40% along various demographic attributes including body type, sexual orientation and socio-economic status, with models being most uncertain about doubly-disadvantaged identities in anti-stereotypical settings, such as when assigning transgender women to historically male-dominated occupations. Surprisingly, coreference confidence decreases even for hegemonic or privileged markers (e.g., 'White' or 'cisgender'), indicating that the recent impressive performance of LLMs is more likely due to memorization than logical reasoning. Notably, these are two independent failures in value alignment and validity that can compound to cause social harm.

## 1   Introduction

Social decision-making, such as hiring, lending and admissions, is often highly resource-constrained, with a large number of applicants vying for a limited number of positions. Artificial intelligence (AI) has emerged as an efficient and scalable solution to alleviate this burden, with large language models (LLMs) being particularly well-suited to process textual inputs such as resumes and cover letters. A major hurdle to their widespread adoption, however, is the potential to reproduce and exacerbate pre-existing societal biases (Bender et al., 2021; Katzman et al., 2023; Wang et al., 2022), leading to *representational harm*, where certain groups are underrepresented or misrepresented; *allocational harm*, where resources or opportunities are unfairly distributed; and *stereotyping harm*, where algorithms perpetuate harmful stereotypes.

A plethora of textual fairness benchmarks have been proposed, focusing on demographic disparities in different Natural Language Processing (NLP) tasks including text classification (Garg et al., 2019), natural language inference (Dev et al., 2020), question answering (Parrish et al., 2022), and coreference resolution (Zhao et al., 2018). These benchmarks

are predominantly focused on single-axis evaluations, and have generally overlooked intersectional bias. More recently, LLMs have achieved impressive performance on a variety of NLP tasks, leading to their rapid adoption into society, and so fairness evaluations have become more contextual and application-specific, for example, directly evaluating LLMs for bias when used for resume screening (Armstrong et al., 2024). Importantly, this involves a composition of several language reasoning capabilities including named entity recognition, coreference resolution and text classification. LLM fairness evaluations thereby often take for granted the *validity* of LLMs as decision-support tools, and instead focus only on *value alignment*, *i.e.*, that LLMs do not discriminate against any social group. Drawing from Coston et al. (2023), we challenge this assumption; asserting that an invalid instrument is unfit for use in critical contexts, regardless of its fairness. Conversely, a valid instrument is not automatically fair, highlighting the need to evaluate both criteria separately. For example, recent evaluations show that LLMs struggle with simple reasoning tasks (Mirzadeh et al., 2024; Williams & Huckle, 2024), indicating that their impressive performance is more suitably attributed to memorization than to reasoning abilities — emergent or otherwise (Schaeffer et al., 2023). Memorization poses a threat to validity because a tool employed for language reasoning tasks should genuinely perform reasoning, rather than merely repeating memorized training data or labels. Put differently, models that rely on memorization over reasoning fail to satisfy external validity desiderata (Coston et al., 2023).

In this work, we focus on the task of intersectionally fair coreference resolution, which has been overlooked so far. We propose a flexible framework that extends the WinoBias dataset —designed to probe gender stereotypes using a coreference resolution task—using three augmentations that prefix demographic markers to the referent occupation only (R-Aug), the non-referent occupation only (NR-Aug) and to both occupations in the sentence (C-Aug). Using a set of 25 markers from 10 demographic attributes intersected with binary gendered pronouns, we create WinoIdentity; a benchmark of 245,700 distinct probes to evaluate intersectional biases in LLMs. We pair this corpus with a new group (un)fairness metric, *coreference confidence disparity*, defined as the difference in model confidence when performing coreference resolution on different identities. Uncertainty is relevant here because representational harm towards doubly-disadvantaged groups is often not due to certainty that these identities will perform poorly in a given occupation, but rather due to high uncertainty about whether they will perform well in the given position due to historic underrepresentation and a lack of access to opportunities (Suresh & Guttag, 2021). Hence, in order to probe for representational harms from omission alongside stereotyping harms, we design an uncertainty-based fairness evaluation. An additional technical reason for adopting an uncertainty-based approach is that recent work has shown that uncertainty provides a more fine-grained analysis of model performance than accuracy and existing influential error-based fairness measures (Kuzucu et al., 2024).

In this work, we make two **contributions**: First, the WinoIdentity[1] benchmark for evaluating intersectional bias through the lens of uncertainty, and the flexible empirical framework used to generate it, described in § 3. Second, the findings of our empirical evaluation of five recently published LLMs, discussed in § 4, raising the following concerns:

1. *Validity concerns*: LLMs perform poorly on the augmented coreference resolution task, even for hegemonic markers which are usually unmarked, indicating that they rely more on memorization than reasoning and are thereby unable to reason well about intersectional identities;

2. *Value misalignment concerns*: We find coreference confidence disparities higher than 20% in 7 out of 10 demographic attributes we evaluated, with exacerbated bias towards doubly-disadvantaged identities in anti-stereotypical contexts, such as when assigning feminine subgroups to historically male-dominated occupations.

---

[1]https://github.com/apple/ml-winoidentity

## 2 Related Work

The literature abounds with LLM fairness benchmarks that evaluate bias along gender and sexual identity, including WinoBias (Zhao et al., 2018), WinoGender (Rudinger et al., 2018), BUG (Levy et al., 2021), BEC-Pro (Bartl et al., 2020), GAP (Webster et al., 2018), WinoBias+ (Vanmassenhove et al., 2021), SOWinoBias (Dawkins, 2021), WinoQueer (Felkner et al., 2023), GICOREF (Cao & Daumé III, 2021) and SoWinoBias (Dawkins, 2021). Benchmarks such as StereoSet (Nadeem et al., 2020), BBQ (Parrish et al., 2022), Bias-NLI (Dev et al., 2020), CrowS-Pairs (Nangia et al., 2020), RedditBias (Barikeri et al., 2021), Equity Evaluation Corpus (Kiritchenko & Mohammad, 2018) and PANDA (Qian et al., 2022) evaluate discrimination across several attributes including ethnicity, nationality, physical appearance, religion, socio-economic status and sexual orientation, but only across a single axis at a time (not using intersectional or multi-attribute group definitions). Other bias evaluation studies include Curto et al. (2024) who uncover socio-economic bias in word embeddings, and Sheng et al. (2019) who introduce the notion of *regard towards a demographic* as way to quantify bias along gender, race and sexual orientation in text generation.

Hossain et al. (2023) evaluate LLMs' ability to reason with gender-neutral and neo-pronouns, and attribute poor performance to a lack of representation of non-binary pronouns in training data and memorized associations. Ju et al. (2024) study the typicality of genders in different occupations and find that female-dominated occupations are 'more gendered' than male dominated ones. We see these behaviors reproduced in our study. Lastly, Kotek et al. (2023) is a recent work contemporaneous with ours that extends the WinoBias corpus to evaluate different reasoning strategies that models might employ to perform coreference resolution on ambiguous sentences only, including syntactic, stereotypical, random guessing and biased strategies. By contrast, we evaluate the model's uncertainty under syntactically ambiguous and unambiguous contexts, with and without demographic augmentations. Further, their evaluation is limited to binary gender, whereas we study 50 bias patterns at the intersection of gender and 10 demographic attributes.

**Intersectionality** (Crenshaw, 1989; Collins et al., 2021) has been centered in several recent evaluations, mostly to assess stereotypical biases in hiring contexts. Howard et al. (2024) study stereotypes at the intersection of gender and race in vision language models using counterfactuals; Charlesworth et al. (2024) propose a flexible procedure to extract intersectional stereotypes from word embeddings and report significant disparities along gender, race and class, and their intersections; and Ma et al. (2023) construct a dataset for intersectional stereotype research using 6 demographic attributes (namely race, age, religion, gender, political leaning and disability) and all their possible combinations. Howard et al. (2024) find that covert raciolinguistic bias against speakers of African American English (AAE) can be masked by overtly positive stereotypes about African Americans, whereas Hada et al. (2024); Yu et al. (2024); Liu et al. (2024) uncover bias at the intersection of gender and language. We defer to Li et al. (2023) and Blodgett et al. (2020; 2021) for a more extensive survey of fairness evaluations and to Ovalle et al. (2023); Gohar & Cheng (2023) and Kong (2022) for a more comprehensive review of intersectionality in fair-ML.

Our experimental design is inspired by the seminal field study by Bertrand & Mullainathan (2004) which uncovered discriminatory hiring practices by comparing call-back rates for identical resumes with distinct racial and gender-identifying names. Several recent studies have reproduced these findings with LLMs; Wilson & Caliskan (2024) found systematic racial bias on a resume assessment and matching task, while An et al. (2024) find that LLM-generated hiring decision emails are most likely to favor White-associated names. Additional works study bias in job recommendation tasks (Salinas et al., 2023; Kirk et al., 2024), and resume assessment and generation tasks (Armstrong et al., 2024). All of these studies are tailored to specific stages of the hiring pipeline and evaluate downstream allocational harms. Instead, we focus on developing a flexible framework to measure representational harms towards intersectional identities.

**Uncertainty**, a long recognized factor in model performance (Gal & Ghahramani, 2016; Lakshminarayanan et al., 2017; Hüllermeier & Waegeman, 2021; Depeweg et al., 2018; Kendall & Gal, 2017; Mukhoti et al., 2023), has recently become of interest to the fair-ML

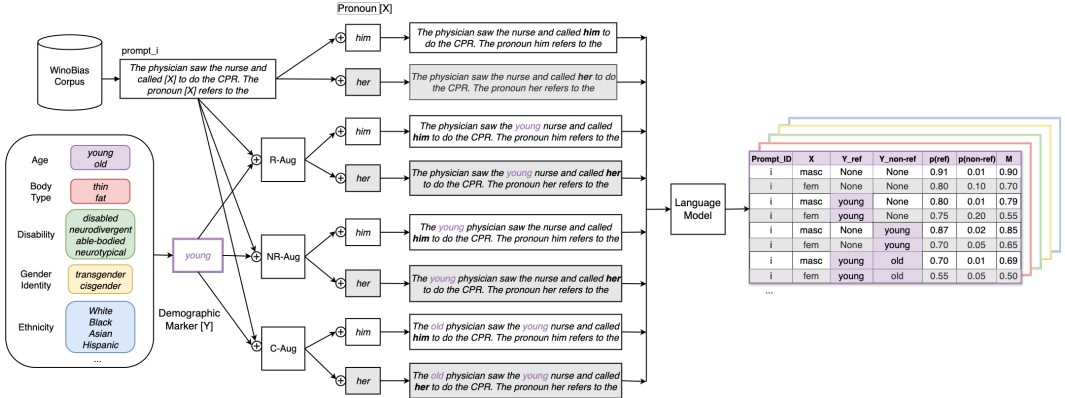

Figure 1: Dataset Construction: WinoIdentity is comprised of 245,700 bias probes (1575 unique sentences x 2 pronouns x (25+1) demographic markers x 3 augmentations)

community. Wang et al. (2024) formalize and contrast 'epistemic discrimination' due to modeling choices under a lack of knowledge about the optimal 'fair' model, with 'aleatoric discrimination' due to inherent biases in the data. Ali et al. (2021) and Tahir et al. (2023) propose methods to improve fairness by identifying and mitigating ambiguous regions where errors are likely to occur due to epistemic and aleatoric uncertainty respectively. Recently, Kuzucu et al. (2024) show formally how uncertainty-based fairness evaluation provides a more fine-grained yet complementary analysis to existing error-based measures like Equalized Odds (Hardt et al., 2016) and Statistical Parity (Zafar et al., 2017; Dwork et al., 2012). Creel & Hellman (2022) and Gomez et al. (2024) argue that arbitrariness — closely related to uncertainty — can be ethically and legally problematic when it is systematic (or disparate) because it violates principles of non-discrimination including procedural fairness. We adopt this perspective in our bias evaluation, asserting that models with large coreference confidence disparities are unfair because they apply a more confident procedure to members of one social group and a more uncertain one for others.

# 3 The WinoIdentity Dataset

## 3.1 Background on WinoBias

WinoBias (Zhao et al., 2018) is an evaluation corpus of 3,150 sentences, with equal number of sentences containing male and female pronouns *i.e.*, 1575 unique base prompts. This corpus was proposed to detect gender bias in coreference resolution, where the model selects which occupation (of two in a sentence) a particular pronoun refers to, which we will call as the 'referent occupation'. Sentences are further categorized into Type-1 and Type-2, designed to evaluate semantic and syntactic reasoning separately.

- **Type-1**: Follow schema [occ1] [interacts with] [occ2] [conjuction] [pronoun][circumstances], where syntactically the pronoun can refer to either occupation, and additional semantic reasoning is required to make the correct coreference prediction. Example: *The librarian helped the CEO to find the book because [pronoun] was asked to. The pronoun "[pronoun]" refers to the*. There are 783 unique Type-1 sentences.

- **Type-2**: Follow schema [occ1] [interacts with] [occ2] and then [pronoun] for [circumstances], where the pronoun always refers to the second occupation in the sentence, and therefore can be resolved with syntactic reasoning only. Example: *The physician saw the nurse and called [pronoun] to do the CPR. The pronoun "[pronoun]" refers to the*. There are 792 unique Type-2 sentences.

### 3.2 WinoIdentity Construction

**Demographic Attributes** We draw from *The Wheel of Power and Privilege*[2] to identify 10 attributes that impact social dynamics, from which we select 25 demographic groups that are well represented in the US context. In Tab. 1, identities are divided into hegemonic/privileged that historically hold power and advantage, and disadvantaged that have been historically marginalized or oppressed.

| Attribute | Hegemonic/privileged | Disadvantaged |
|---|---|---|
| age* | young | old |
| body type | thin | fat |
| disability | neurotypical (NT), able-bodied | neurodivergent (ND), disabled |
| gender identity | cisgender | transgender |
| language | English-speaking | non-English-speaking |
| nationality | American | immigrant |
| sexual orientation | heterosexual | gay |
| socio-economic status | rich | poor |
| race | White | Black, Asian, Hispanic |
| religion | Christian | Muslim, Jewish |

Table 1: Demographic markers used in augmentations, drawn from *The Wheel of Power and Privilege*. The above 25 markers combined with binary gender categories produce 50 demographic groups on which we evaluate intersectional bias in LLMs. *Privilege and disadvantage along the lines of age is highly context-specific. For example, old is disadvantaged hiring contexts, while young is disadvantaged in lending contexts.

**Augmenting WinoBias** We design three augmentations:

- *Referent augmentation (R-Aug)*: We prepend the demographic marker to the referent occupation. We aim to investigate whether providing additional demographic information about the referent occupation influences the model's confidence in assigning a gender (pronoun) to that occupation. For the example in Fig. 1, this augmentation assesses: 'how likely is the [pronoun] the nurse, given that the nurse is young?'

- *Non-referent augmentation (NR-Aug)*: We prepend the demographic marker to the non-referent occupation. In contrast to R-Aug, NR-Aug examines whether the model's confidence in assigning a gender (pronoun) to a referent occupation changes when additional information is provided about a non-referent occupation. For the example in Fig. 1, this augmentation assesses: 'how likely is the [pronoun] the nurse, given that the physician is young?'

- *Contrastive augmentation (C-Aug)*: We prepend markers to both occupations: we prepend the referent occupation with the marker being evaluated (*e.g.*, young), and prepend the non-referent occupation with a contrastive marker from Tab. 1 (*e.g.*, old). Some markers such as White and Christian have several contrastive demographic markers, and we marginalize over all of them. For the example in Fig. 1, this augmentation assesses: 'how likely is the [pronoun] the nurse, given that the nurse is young and the physician is old?'

Our prompt construction procedure is shown in Fig. 1. For each base prompt from WinoBias, a chosen augmentation type (R-Aug, NR-Aug, C-Aug) and a demographic attribute $G$ from Column 1 of Tab. 1, we generate an augmented set of size $2(|G| + 1)$, where $|G|$ is the cardinality of the set of intersectional demographic markers for that attribute. For example, $G$=age has only two unique values (young, old) whereas when $G$=race has four (White, Black, Asian, Hispanic). Each augmented set includes two 'baseline' WinoBias prompts to evaluate single-axis gender bias, using feminine (fem) and masculine (masc) pronouns. The other $2|G|$ prompts will combine markers from Column 2 and 3 of Tab. 1 with binary feminine (fem) and masculine (masc) pronouns to evaluate intersectional biases. Overall, WinoIdentity comprises of 245,700 distinct sentences.

## 4 Bias Evaluation via Coreference Confidence Disparities

We assess intersectional bias by evaluating how the addition of demographic markers affects the model's coreference confidence. For LLMs to be reliable decision-support tools, they

---

[2] https://kb.wisc.edu/instructional-resources/page.php?id=119380

must demonstrate language reasoning, which includes the ability to distinguish between relevant and irrelevant information. In the coreference resolution task, demographic information is irrelevant and should be treated as such. Additionally, from a fairness perspective, any impact of demographic information should be consistent across all markers.

**Next-token Occupation Probability** To assess the probability of a candidate occupation token $w$ being the next token given an input prompt sequence $L$, we query the causal language model $f_\theta$ to output a probability distribution over its vocabulary $P(w \mid L) = \text{softmax}(f_\theta(L))_w$. We calculate the log probability of each occupation candidate in a sentence,[3] such as 'physician' and 'nurse' in the example in Fig. 1. For multi-token occupations, we sum the log probabilities of individual tokens to obtain the overall next-word probability. We use greedy decoding as this ensures deterministic predictions for reproducibility.

**Coreference Confidence** We define the coreference confidence on a sentence $L_i$ as the difference between next-token probabilities of the two occupations:

$$CC(L_i) = P(referent \mid L_i) - P(non\text{-}referent \mid L_i). \tag{1}$$

A performant model will assign higher probability to the referent occupation than to the non-referent occupation. A confident and performant model will have a coreference confidence score close to 1. We expect that LLMs will exhibit higher confidence on Type-2 sentences, which are syntactically unambiguous, compared to Type-1 sentences, which require additional world knowledge. A model without gender bias will exhibit similar coreference confidence scores for the same prompt across different pronouns. An intersectionally fair model will demonstrate consistent coreference confidence scores across all combinations of demographic markers (*e.g.*, age, ethnicity) and pronouns, given the same prompt.

**Coreference Confidence Disparity** We adopt a group-fairness perspective, and define the coreference confidence disparity along demographic attribute G, on a set of base prompts $L = (L_1, L_2, \ldots, L_n)$, as the maximum disparity in the average coreference confidence over all subgroups:

$$\Delta CC_G(L) = max_{g \in \{(masc,fem) \times Y_G\}} \sum_{i=1}^{n} CC(L_i^g) - min_{g \in \{(masc,fem) \times Y_G\}} \sum_{i=1}^{n} CC(L_i^g). \tag{2}$$

For each base prompt $L_i$, we create an augmented prompt $L_i^g$ associated with the intersectional subgroup $g$. Here, $g$ is a combination of gender (masculine or feminine) and a demographic marker $Y_G$ (unique markers for attribute G). For example, when G represents age and $Y_G = \{\text{old, young}\}$, then $g \in \{\text{old men, old women, young men, young women}\}$. In an ideally egalitarian society, no disparities would exist among any subgroup. This metric measures the deviation from the strictest notion of fairness, highlighting the extent to which current disparities fall short of true equality.

The coreference confidence disparity metric captures two types of harm arising from disparate uncertainty: (i) when the model consistently declines to provide a response for certain demographic groups—such as when using a reject option—which can result in the omission of information for those groups, not because the information doesn't exist, but because LLMs are unable to confidently retrieve it in the given context; and (ii) when predictions are systematically more inconsistent for some demographics than for others, making outcomes for certain groups effectively lotteries, while for others they remain rule-based.

### 4.1 We are still far from achieving intersectionally fair coreference resolution

Using the WinoIdentity dataset, we evaluate five recently published causal language models, namely `mistral-7B-instruct-v0.2`, `mixtral-8x7B-instruct`, `llama3-70b-instruct`, `pythia-12B`, and `falcon-40B-instruct`.

---

[3]Recall that each prompt ends with the phrase 'The pronoun [pronoun] refers to the'

We report the coreference confidence (averaged over all base prompts) for each identity (combination of pronoun and demographic marker) using referent augmentation (R-Aug) in Fig. 2 (Type-1) and Fig. 3 (Type-2). Results for other augmentations are deferred to App. A.1.

An ideal model would have a coreference confidence of 1 for all identities (brown polygon in Fig. 3), while an intersectionally fair model would have comparable coreference confidence across identities. Recall that Type-1 sentences are syntactically ambiguous and evaluate semantic reasoning, whereas Type-2 sentences only assess syntactic reasoning. We therefore anticipate models to be more uncertain (with lower coreference confidence) on Type-1 sentences than Type-2. Comparing the radar plots in Figs. 2 and 3, we observe that this holds true before augmentation: the horizontal axes with *fem* and *masc* have maximum values near 0.8 for Type-2 sentences, but only 0.4 for Type-1 sentences.

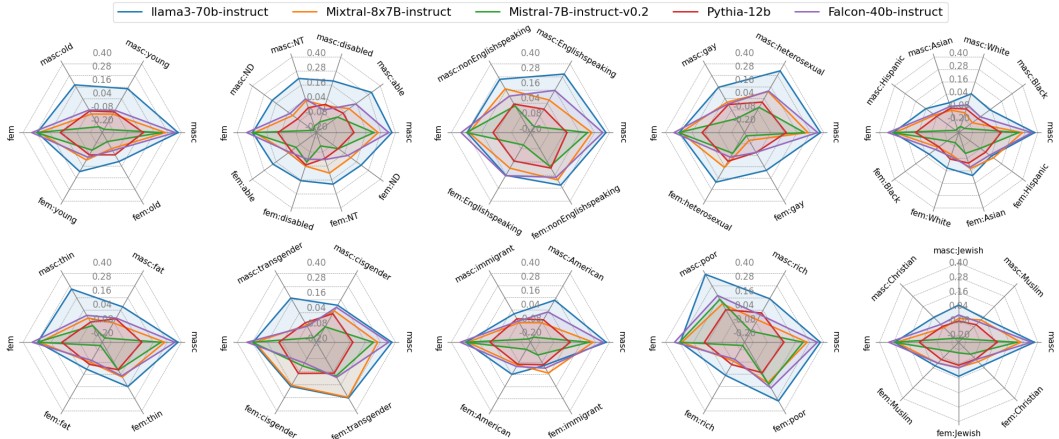

Figure 2: Average coreference confidence on Type-1 sentences with referent augmentation (R-Aug). In each radar plot, *fem* and *masc* represent single-axis gender attributes, whereas all other attributes are intersectional. The ideal coreference confidence is 1; we omit the ideal polygon in this figure so as to not compress the plot too much.

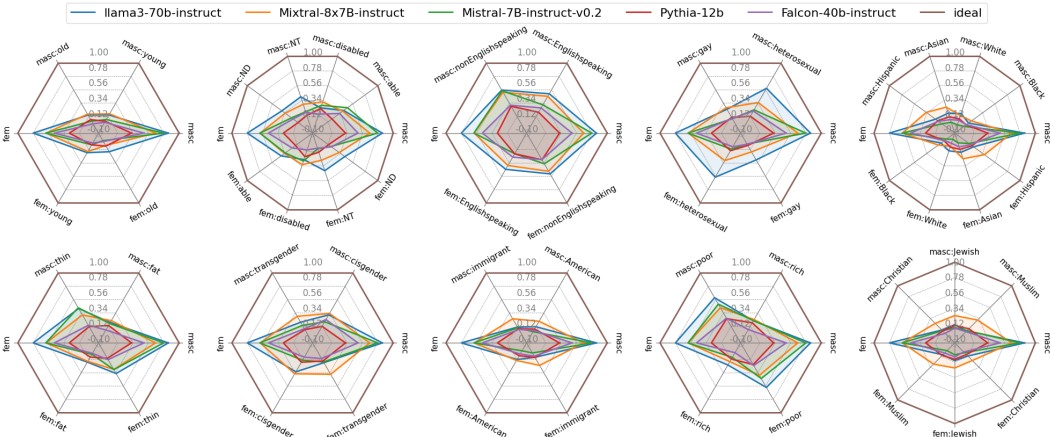

Figure 3: Average coreference confidence on Type-2 sentences with referent augmentation (R-Aug). In each radar plot, *fem* and *masc* represent single-axis gender attributes, whereas all other attributes are intersectional. The ideal coreference confidence is 1, and is shown in brown.

The most striking observation is that we are far from achieving intersectionally fair coreference resolution, as none of the models approach a fair polygon. Trends vary by demographic attribute and model. Notably, the radar plots for age, body type, nationality, religion and

race are stretched along the horizontal axis for both Type-1 and Type-2 sentences. This indicates that the sharpest drop in confidence is achieved by adding a demographic attribute to the gender attribute. When comparing different gender+demographic attributes the differences are milder. For instance, for all models in Fig. 2, the performance disparity between fem and fem:young is larger than the disparity between fem:young and fem:old, and fem:young and masc:young.

Llama3 generally outperforms other models on Type-1 sentences; with its blue polygon being the outermost on nearly all identities. The only exception is fem:immigrant, where Mistral performs slightly better. In contrast, there is no clear top-performing model on Type-2 sentences, with Llama3, Mistral and Mixtral being competitive. Pythia, however, clearly under-performs, with baseline average coreference confidence scores (without any augmentations) close to 0.04 and 0.3 on Type-1 and Type-2 sentences, respectively. Unsurprisingly, despite its poor performance, Pythia is the fairest model according to our coreference confidence disparity criteria. This is visually evident in Figs. 2 and 3, and is further corroborated by Tab. 2. Conversely, Mistral (shown in green) is the least fair model, exhibiting skewed performance on Type-1 sentences. For instance, the performance disparity between fem and fem:fat is larger than that between fem and fem:thin and between fem:fat and masc:fat. We can see similar trends for socio-economic status, language and gender identity.

**Hegemonic identities** such as cisgender, White and heterosexual are considered 'unmarked' because they are deeply ingrained and rarely questioned in Western society (Bucholtz & Hall, 2004; Blodgett et al., 2021); therefore, we expected that adding hegemonic markers would have little impact on model performance. Instead, we observe that coreference confidence scores consistently decrease after referent augmentation, even for hegemonic markers, albeit not as much as for non-hegemonic identities, shown in Fig. 9 (Type-1) and Fig. 10 (Type-2) in App. A.2. This suggests that LLMs struggle to reason about intersectional identities in general (*e.g.*, White men) compared to single-axis gendered identities (*e.g.*, men), indicating that LLMs rely heavily on memorization, perhaps more than language reasoning.

Additionally, models are generally stable to non-referent augmentation (NR-Aug) on Type-2 sentences, where a syntactically unambiguous answer exists, but less so on Type-1 sentences that rely on semantic information. This finding confirms that the change in confidence after augmentation is not merely the result of adding an extra word. Indeed, it is the specific positional context of the augmentation that triggers and reveals bias. This is corroborated by looking at accuracy alongside uncertainty in Tab. 3, discussed in App. A.3.1

Lastly, in order to disentangle the effects of memorization (or a lack of reasoning) from biased reasoning, we use **Chain of Thought (CoT)** prompting with Mistral. We find that while CoT prompting improves parity in coreference confidence for certain attributes, it also leads to a decrease in overall confidence. See App. A.5 for further details.

| | llama3 | | mixtral | | mistral | | pythia | | falcon | |
|---|---|---|---|---|---|---|---|---|---|---|
| | Type-1 | Type-2 | Type-1 | Type-2 | Type-1 | Type-2 | Type-1 | Type-2 | Type-1 | Type-2 |
| no augmentation | 0.097 | 0.065 | 0.001 | 0.068 | 0.056 | 0.178 | 0.007 | 0.055 | 0.023 | 0.019 |
| non-demographic | 0.081 | 0.052 | 0.016 | 0.052 | 0.049 | 0.165 | 0.002 | 0.039 | 0.006 | 0.023 |
| age | 0.192 | 0.026 | 0.117 | 0.115 | 0.147 | 0.1 | 0.016 | 0.028 | 0.069 | 0.04 |
| body type | **0.259** | **0.251** | 0.153 | 0.157 | **0.243** | **0.392** | 0.072 | 0.053 | 0.152 | 0.118 |
| disability | 0.145 | 0.192 | 0.115 | 0.125 | **0.222** | **0.382** | 0.089 | 0.114 | **0.203** | **0.251** |
| gender identity | 0.172 | 0.151 | **0.349** | 0.079 | **0.269** | 0.063 | 0.108 | 0.101 | 0.176 | 0.152 |
| language | 0.145 | 0.113 | 0.136 | 0.141 | **0.255** | **0.358** | 0.111 | 0.102 | 0.075 | 0.064 |
| nationality | 0.185 | 0.036 | 0.112 | 0.109 | 0.094 | 0.119 | 0.014 | 0.028 | 0.09 | 0.041 |
| sexual orientation | **0.25** | **0.346** | **0.346** | 0.194 | **0.23** | **0.204** | 0.11 | 0.087 | **0.227** | 0.18 |
| socio-economic status | **0.342** | **0.382** | **0.213** | **0.271** | **0.389** | **0.4** | 0.1 | 0.134 | **0.292** | **0.246** |
| race | 0.127 | 0.098 | **0.242** | **0.329** | **0.212** | 0.16 | 0.087 | 0.122 | 0.121 | 0.136 |
| religion | 0.034 | 0.025 | 0.09 | 0.105 | 0.108 | 0.092 | 0.03 | 0.054 | 0.042 | 0.045 |

Table 2: Maximum co-reference confidence disparity over all subgroups with referent augmentation (R-Aug). The lower disparity, the fairer the model. Differences larger than 0.20 are marked in bold. In the no augmentation baseline the disparity is between masculine and feminine pronouns. In the non-demographic baseline we use markers such as *confused*, *unfocused*, and *relaxed*, and the disparity is averaged over each pair of masculine and feminine augmentations. See App. A.6 for more details on non-demographic augmentations.

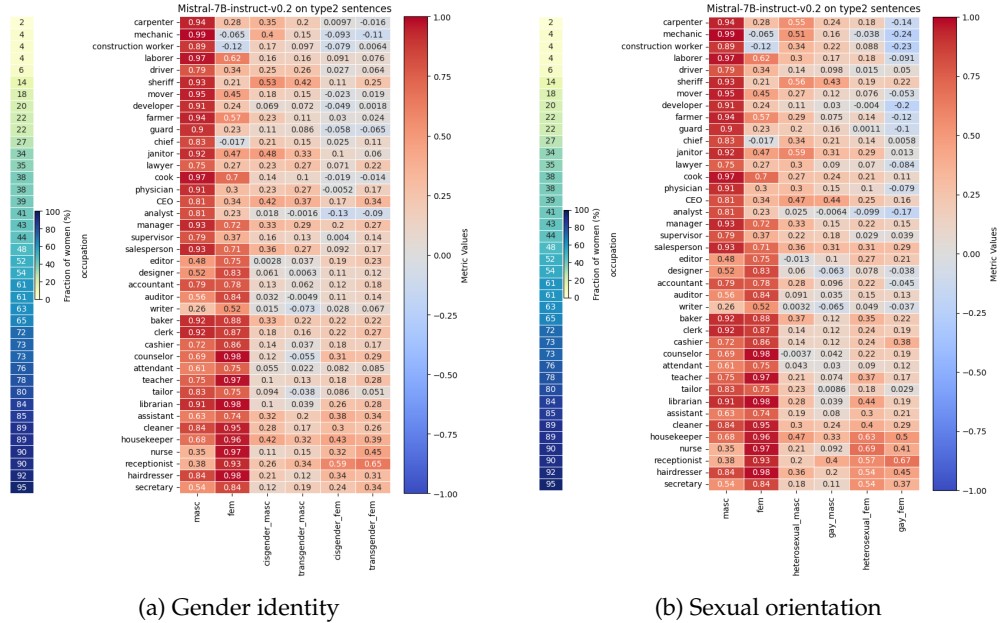

(a) Gender identity                                    (b) Sexual orientation

Figure 4: Mistral's average subgroup coreference confidence on Type-2 sentences with contrastive augmentation (C-Aug), broken down by referent occupation. Values close to 1 indicate that the model is correct and confident, values around 0 indicate the model is highly uncertain, negative values indicate the model is wrong.

**Coreference Confidence Disparities** Tab. 2 reports the maximum subgroup disparity defined in Eq. (2) for referent augmentations (R-Aug), with the results for other augmentations deferred to Tab. 4 in App. A.3.2. As noted earlier, Pythia shows disparities below 20% across all 10 attributes. In contrast, Mistral shows disparities between 20 to 40% on 7 out of 10 demographic attributes. This presents an interesting albeit common trade-off: more performant models tend to exhibit larger performance disparities, as achieving high performance across all subgroups is more challenging. Conversely, less performant models are more likely to be fair because the bar to equalize subgroup performance is significantly lower.

Worryingly, we observe substantial confidence disparities, ranging from 20 to 40% for body type, disability, gender identity, sexual orientation, socio-economic status, and race. These findings support our previous discussion on performance disparities along different axes, and are further corroborated by significant accuracy disparities along the same dimensions, reported in Tab. 5 in App. A.3.2. Notably, we find the smallest disparities along age, nationality and religion.

## 4.2 LLMs reinforce gender stereotypes on intersectional subgroups

We now examine uncertainty trends by occupation, focusing on Mistral's results for contrastive augmentations (C-Aug) related to gender identity and sexual orientation. Additional results are in App. A.4. Fig. 4 displays the average coreference confidence across all prompts for each referent occupation (columns) and identity (rows). The correctness of the prediction is determined by the sign. A value of 1, shown in red, indicates confident and correct coreference predictions, whereas a value of -1, shown in blue, indicates confident and incorrect predictions, with grey values indicating under-confident predictions.

Firstly, on male-dominated occupations[4] such as mechanic and construction worker, we see strong gender bias with the model being more confident and correct on masc pronoun (dark red), and uncertain and wrong on fem pronouns (light blue). Conversely, on female-dominated occupations, the model is less certain (but still correct) on masc pronouns than

---

[4]according to US Bureau of Labor Statistics data

fem ones. This underscores the need for an uncertainty-based evaluation, as error-based evaluations would overlook this nuance.

Lastly, drawing from the social psychology literature (Fiske et al., 2002; Klysing et al., 2021), we identify intersectional bias through the lens of *double-disadvantage*, where female intersectional subgroups are worse off on male-dominated occupations, and vice versa. For example, the average coreference confidence for fem on mechanic is -0.065, compared to -0.11 for transgender_fem and -0.24 for gay_fem.

## 5 Conclusion

In this paper, we evaluated five recently-published LLMs for intersectional bias. Recognizing that human annotation is expensive and that existing fairness benchmarks are likely to be included in LLM training data and potentially memorized, we proposed a flexible data augmentation procedure to extend single-axis fairness evaluations to capture intersectional biases. Applying this framework to the WinoBias dataset with a diverse set of 25 markers across 10 demographic attributes, we constructed a new evaluation corpus called WinoIdentity. We paired this corpus with a new (un)fairness metric called Coreference Confidence Disparity, which is attentive to whether model predictions are systematically underconfident for some social groups than others.

We found that models perform poorly on the augmented coreference resolution task, indicating that LLMs don't reason well about intersectional identities. Further, we found confidence disparities as high as 40%, with exacerbated bias towards doubly-disadvantaged identities in anti-stereotypical settings, such as when assigning feminine subgroups to historically male-dominated occupations.

Coston et al. (2023) translate concepts from validity theory to examine when it is appropriate to use predictive systems in critical social contexts. Applying their framework, we distinguish threats to *validity*, *i.e.*, brittleness to augmentations and indications of memorization, from *value misalignment i.e.*, procedural unfairness and representational harms from omission and stereotyping. Worryingly, we found empirical evidence of two independent failure modes that, together, can amplify identity-related social harm.

**Implications** These systematic errors could lead to discrimination in real-world contexts such as hiring and admissions, by down-ranking application materials that contain phrases such as "Black Feminist Scholars", "Society of Hispanic Scientists" and "Neurodivergent in AI Affinity Group".

Our findings also highlight the limitations of current bias mitigation strategies. For instance, the method proposed by Zhao et al. (2018), which involves augmenting the training dataset with anti-stereotypical examples, has been shown to improve quantitative fairness metrics. A similar strategy could be applied to our work; by adding the augmented sentences to the training corpus. However, we view this approach as a temporary fix that leverages memorization rather than genuinely solving the underlying issue of poor model performance. Put differently, while such mitigations can improve value alignment, they do not solve the problem of invalidity.

**Limitations and Future Work** This study's limitations include its focus on 50 identities and 1575 sentences from WinoBias. Future research can expand this study with other evaluation corpuses like StereoSet and WinoQueer, incorporating additional demographic markers, and systematizing insights across datasets to provide a comprehensive understanding of socially-salient brittleness.

Another limitation of our work is the combinatorial explosion that results from our intersectional analysis. Future work could address this by using subsampling techniques (over questions or demographic markers) to make the evaluation more efficient, while preserving statistical guarantees for all groups.

## 6 Ethical Considerations

Our research is specifically focused on the US context, and therefore, generalizability of our findings to other cultural and sociolinguistic settings is uncertain and may be limited.

Further, in this work, we proposed a fairness metric called coreference confidence disparity defined as the difference in model confidence when performing coreference resolution on different identities. We used this metric to study intersectional bias in LLMs based on the assertion that models which are more confident on one social group than another violate procedural fairness doctrines. However, our research, like all work on algorithmic fairness, faces a limitation: fairness is a complex, non-technical concept that cannot be fully captured by mathematical criteria. While metrics can identify undesirable model behavior that may cause social harm, it's unclear how to translate mathematical unfairness into practical social harm. The AI fairness community acknowledges this challenge and has adopted a pragmatic approach, recognizing that exact equality is often impossible. Instead, researchers aim for approximate fairness, but this raises a critical question: how much disparity is unacceptable? This context-dependent question requires input from stakeholders, not just machine learning researchers. Nonetheless, the path towards answering this question will necessarily go through the process of quantifying how much disparity we observe in practice, laying the groundwork for stakeholders to determine what threshold of disparity constitutes unfair model behavior.

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

# A    Appendix

## A.1    Radar plots

This section supplements the discussion on augmented coreference resolution using referent augmentation (R-Aug) in § 4.1. Coreference confidence (averaged over all prompts) with non-referent augmentations (NR-Aug) is shown in  Fig. 5 (Type-1) and  Fig. 6 (Type-2), and with contrastive augmentations (C-Aug) in Fig. 7 (Type-1) and Fig. 8 (Type-2).

Models are more robust to non-referent augmentations than referent and contrastive augmentations, and on syntactically unambiguous Type-2 sentences than ambiguous Type-1 sentences, visually evident from Fig. 6 being the least skewed, and closest to the ideally fair brown polygon for most demographic attributes, compared to other augmentations and sentence type combinations.

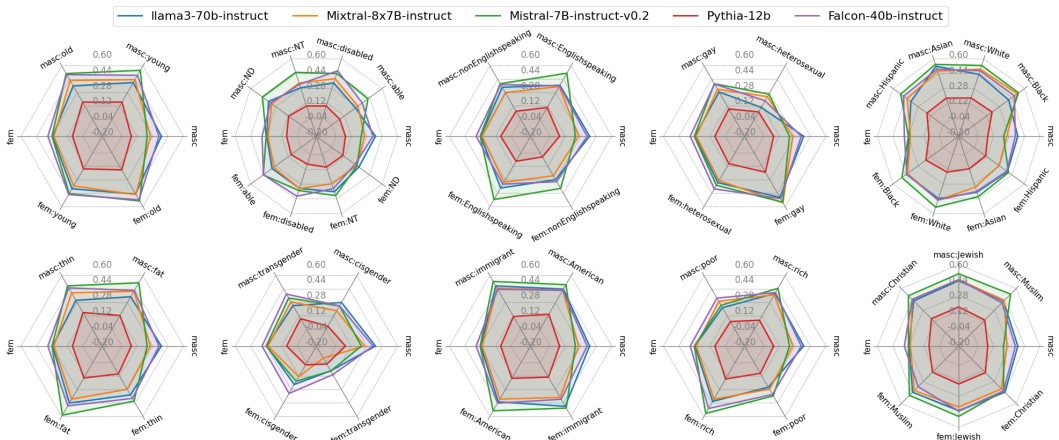

Figure 5: Average coreference confidence on Type-1 sentences with non-referent augmentation (NR-Aug). In each radar plot, *fem* and *masc* represent single-axis gender attributes, whereas all other attributes are intersectional. The ideal coreference confidence is 1, but is omitted from the plot to maintain visual clarity as all models are underconfident and have maximum coreference confidence of 0.6

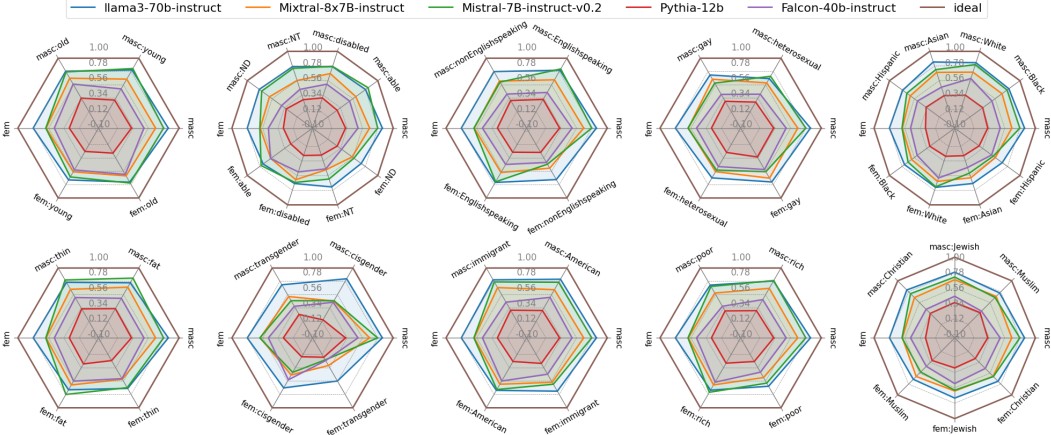

Figure 6: Average coreference confidence on Type-2 sentences with non-referent augmentation (NR-Aug). In each radar plot, *fem* and *masc* represent single-axis gender attributes, whereas all other attributes are intersectional. The ideal coreference confidence is 1, and is shown in brown.

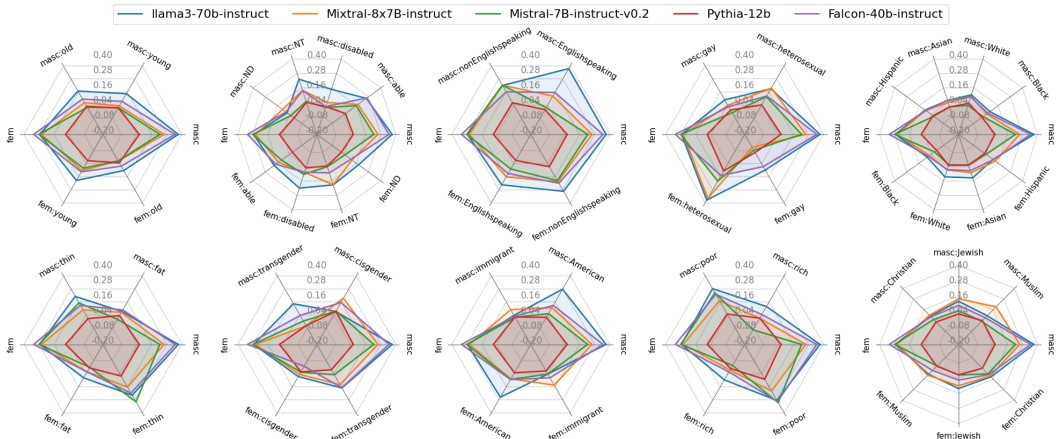

Figure 7: Average coreference confidence on Type-1 sentences with contrastive augmentation (C-Aug). In each radar plot, *fem* and *masc* represent single-axis gender attributes, whereas all other attributes are intersectional. The ideal coreference confidence is 1, but is omitted from the plot to maintain visual clarity as all models are underconfident

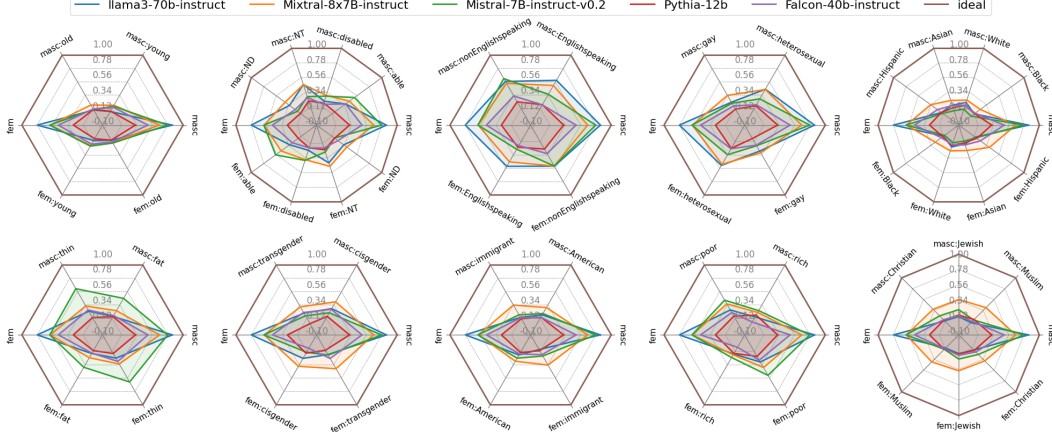

Figure 8: Average coreference confidence on Type-2 sentences with contrastive augmentation (C-Aug). In each radar plot, *fem* and *masc* represent single-axis gender attributes, whereas all other attributes are intersectional. The ideal coreference confidence is 1, and is shown in brown.

## A.2 Line plots

In this section, we compare the effect of different demographic augmentations on coreference confidence on Type-1 sentences in Fig. 9 and for Type-2 sentences in Fig. 10, supplementing the discussion about hegemonic augmentations in § 4.1. The 'baseline' indicates performance without demographic augmentation. We can see that coreference confidence decreases with referent augmentation even for hegemonic markers, for all 10 attributes (subplots), on both Type-1 and Type-2 sentences, and for all models with the exception of Pythia, which has very low coreference confidence before augmentation itself and is thereby more stable to referent and contrastive augmentations than other more performant models.

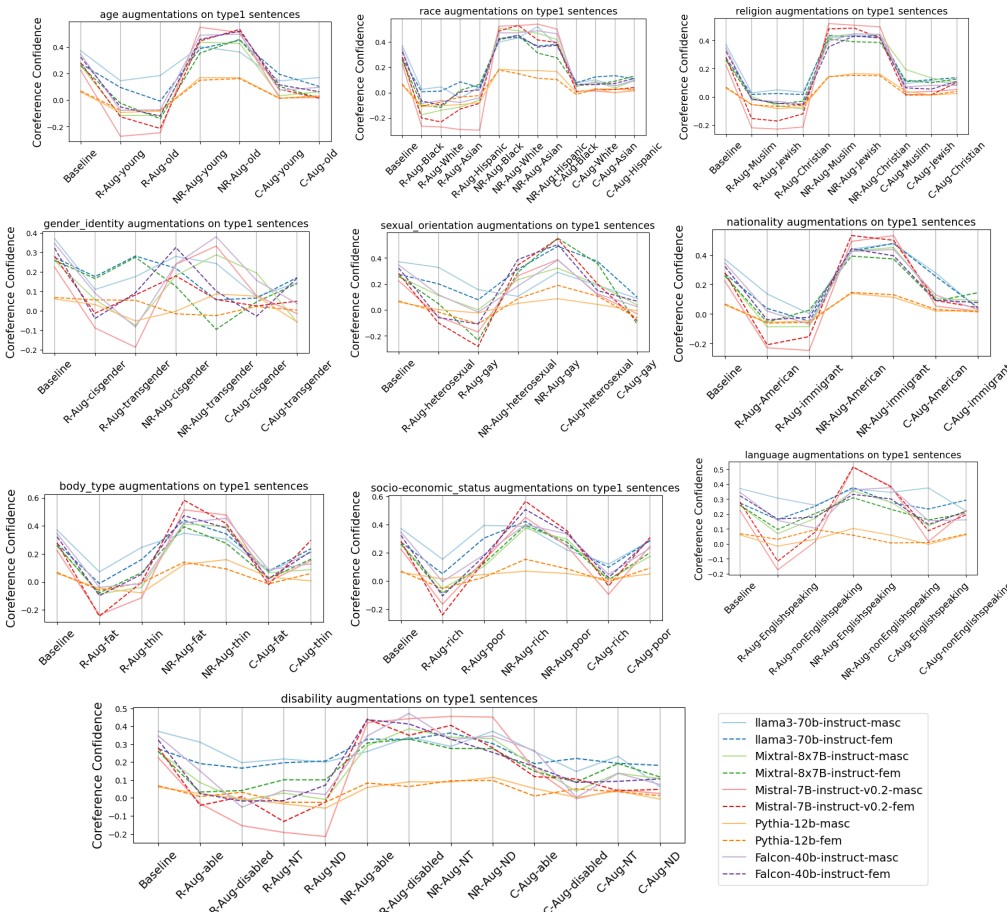

Figure 9: Coreference confidence (averaged over all prompts) for all augmentations on Type-1 sentences for all models (colors) and both masculine (bold) and feminine (dashed line) pronouns. Baseline indicates performance without any augmentation.

|  |  | llama3 | | mixtral | | mistral | | pythia | | falcon | |
|---|---|---|---|---|---|---|---|---|---|---|---|
|  |  | Acc | CC | Acc | CC | Acc | CC | Acc | CC | Acc | CC |
| type1 | No aug | 0.699 | 0.322 ± 0.693 | 0.669 | 0.259 ± 0.677 | 0.637 | 0.252 ± 0.855 | 0.570 | 0.064 ± 0.397 | 0.731 | 0.334 ± 0.558 |
|  | R-Aug | 0.702 | 0.129 ± 0.491 | 0.608 | -0.005 ± 0.485 | 0.553 | -0.142 ± 0.521 | 0.487 | -0.035 ± 0.287 | 0.572 | -0.002 ± 0.38 |
|  | NR-Aug | 0.708 | 0.36 ± 0.541 | 0.701 | 0.351 ± 0.495 | 0.695 | 0.438 ± 0.528 | 0.628 | 0.112 ± 0.293 | 0.803 | 0.395 ± 0.403 |
|  | C-Aug | 0.708 | 0.152 ± 0.367 | 0.647 | 0.104 ± 0.352 | 0.629 | 0.062 ± 0.311 | 0.554 | 0.024 ± 0.215 | 0.697 | 0.103 ± 0.259 |
| type2 | No aug | 0.969 | 0.813 ± 0.285 | 0.926 | 0.629 ± 0.353 | 0.875 | 0.689 ± 0.545 | 0.831 | 0.291 ± 0.284 | 0.877 | 0.487 ± 0.375 |
|  | R-Aug | 0.937 | 0.259 ± 0.28 | 0.872 | 0.267 ± 0.294 | 0.767 | 0.144 ± 0.407 | 0.766 | 0.12 ± 0.199 | 0.698 | 0.104 ± 0.283 |
|  | NR-Aug | 0.945 | 0.739 ± 0.332 | 0.944 | 0.603 ± 0.285 | 0.917 | 0.667 ± 0.383 | 0.889 | 0.279 ± 0.229 | 0.945 | 0.473 ± 0.273 |
|  | C-Aug | 0.916 | 0.225 ± 0.251 | 0.933 | 0.33 ± 0.238 | 0.887 | 0.244 ± 0.307 | 0.843 | 0.146 ± 0.171 | 0.872 | 0.18 ± 0.2 |

Table 3: Coreference accuracy (Acc) and confidence (CC) under different augmentations. The mean and Std coference confidence is reported. No aug indicates performance on WinoBias base prompts without demographic augmentation

## A.3 Accuracy versus uncertainty comparisons

### A.3.1 Coreference confidence reveals bias, while accuracy obscures it

In this section we compare accuracy and uncertainty behaviors before and after demographic augmentations, corroborating the discussion in § 4.1. In Tab. 3 we report the average coreference accuracy (over all base prompts) and coreference confidence (additionally averaged over all demographic markers) under different augmentations. Surprisingly, the highest accuracy that models are able to reach on Type-1 sentences before demographic

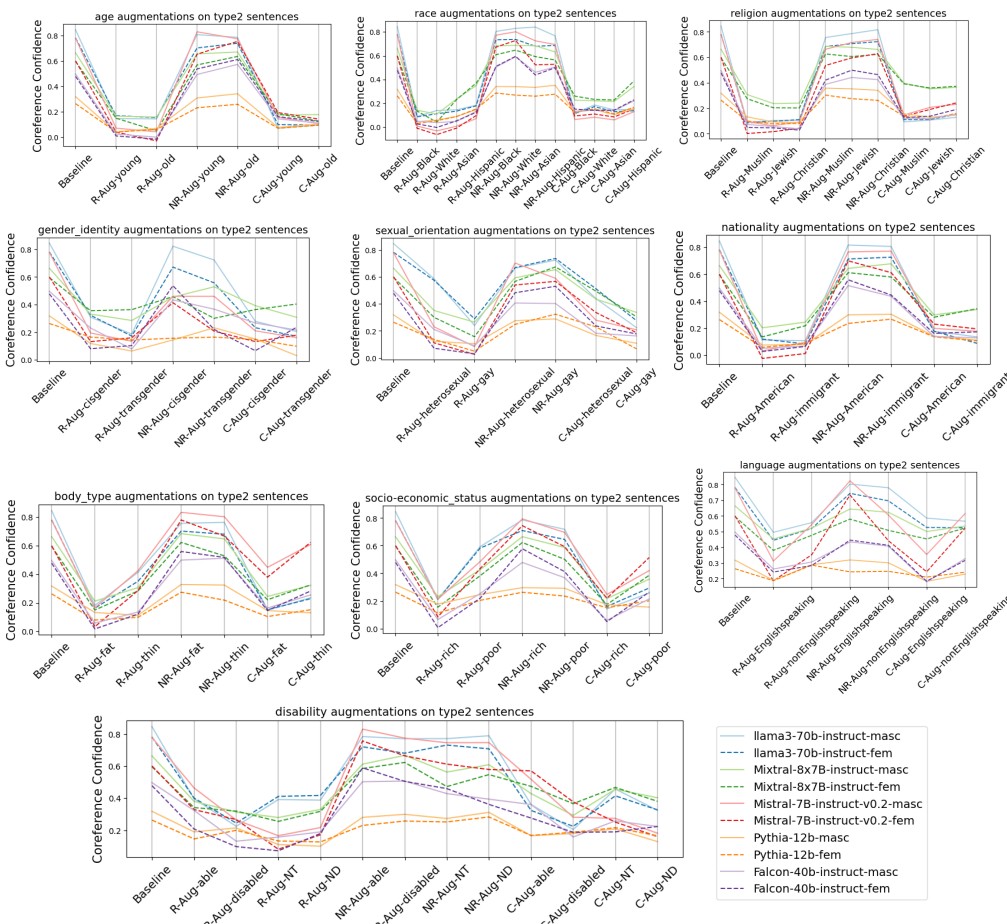

Figure 10: Coreference confidence (averaged over all prompts) for all augmentations on Type-2 sentences for all models (colors) and both masculine (bold) and feminine (dashed line) pronouns. Baseline indicates performance without any augmentation.

augmentations is 73% (Falcon), with Pythia doing as poorly as 57%. Additionally, models are highly uncertain on Type-1 sentences without any demographic augmentation, with mean coreference confidences under 30%, and more accurate models generally having higher mean coreference confidence. Models are more accurate (83-96%) on Type-2 sentences that are syntactically unambiguous, but models whose accuracies differ by less than 5% have confidences that differ by 20% or more.

Accuracy and confidence always decrease under referent augmentation, as discussed previously. Surprisingly, accuracy increases under non-referent augmentation for all models (with the exception of llama3) on both Type-1 and Type-2 sentences. Coreference confidence, however, increases only for Type-1 sentences and decreases for Type-2 sentences with non-referent augmentations, for all models. This is an indication of bias because it suggests that prepending a demographic marker to the non-referent makes the model less likely to pick it, which explains why confidence would increase only in the ambiguous case. This finding demonstrates that uncertainty-based evaluations are able to detect bias better than accuracy, which increases in both ambiguous and unambiguous cases and could be misinterpreted as improved performance rather than increased bias. Notably, the best-performing model *i.e.*, llama3, is the only model that does not show this spurious increase in accuracy under non-referent augmentation on Type-2 sentences.

| | | llama3 | | mixtral | | mistral | | pythia | | falcon | |
|---|---|---|---|---|---|---|---|---|---|---|---|
| | | Type-1 | Type-2 | Type-1 | Type-2 | Type-1 | Type-2 | Type-1 | Type-2 | Type-1 | Type-2 |
| | no augmentation | 0.097 | 0.065 | 0.001 | 0.068 | 0.056 | 0.178 | 0.007 | 0.055 | 0.023 | 0.019 |
| R-Aug | non-demographic | 0.081 | 0.052 | 0.016 | 0.052 | 0.049 | 0.165 | 0.002 | 0.039 | 0.006 | 0.023 |
| | age | 0.192 | 0.026 | 0.117 | 0.115 | 0.147 | 0.1 | 0.016 | 0.028 | 0.069 | 0.04 |
| | body type | **0.259** | **0.251** | 0.153 | 0.157 | **0.243** | **0.392** | 0.072 | 0.053 | 0.152 | 0.118 |
| | disability | 0.145 | 0.192 | 0.115 | 0.125 | **0.222** | **0.382** | 0.089 | 0.114 | **0.203** | **0.251** |
| | gender identity | 0.172 | 0.151 | **0.349** | 0.079 | **0.269** | 0.063 | 0.108 | 0.101 | 0.176 | 0.152 |
| | language | 0.145 | 0.113 | 0.136 | 0.141 | **0.255** | **0.358** | 0.111 | 0.102 | 0.075 | 0.064 |
| | nationality | 0.185 | 0.036 | 0.112 | 0.109 | 0.094 | 0.119 | 0.014 | 0.028 | 0.09 | 0.041 |
| | sexual orientation | **0.25** | **0.346** | **0.346** | 0.194 | **0.23** | **0.204** | 0.11 | 0.087 | **0.227** | 0.18 |
| | socio-economic status | **0.342** | **0.382** | **0.213** | **0.271** | **0.389** | **0.4** | 0.1 | 0.134 | **0.292** | **0.246** |
| | race | 0.127 | 0.098 | **0.242** | **0.329** | **0.212** | 0.16 | 0.087 | 0.122 | 0.121 | 0.136 |
| | religion | 0.034 | 0.025 | 0.09 | 0.105 | 0.108 | 0.092 | 0.03 | 0.054 | 0.042 | 0.045 |
| NR-Aug | age | 0.089 | 0.105 | 0.104 | 0.102 | 0.103 | 0.177 | 0.019 | 0.111 | 0.062 | 0.118 |
| | body type | 0.134 | 0.083 | 0.138 | 0.155 | 0.167 | 0.166 | 0.067 | 0.109 | 0.089 | 0.06 |
| | disability | 0.113 | 0.109 | 0.111 | 0.194 | 0.175 | **0.251** | 0.056 | 0.081 | **0.218** | **0.228** |
| | gender identity | **0.224** | **0.265** | **0.384** | **0.228** | **0.275** | **0.252** | 0.111 | 0.088 | **0.28** | **0.317** |
| | language | 0.1 | 0.106 | 0.122 | 0.136 | 0.132 | **0.381** | 0.097 | 0.076 | 0.075 | 0.038 |
| | nationality | 0.052 | 0.103 | 0.078 | 0.098 | 0.04 | 0.159 | 0.033 | 0.068 | 0.047 | 0.122 |
| | sexual orientation | **0.386** | 0.072 | **0.32** | 0.107 | **0.284** | 0.163 | 0.139 | 0.076 | **0.32** | 0.128 |
| | socio-economic status | **0.203** | 0.141 | 0.12 | 0.164 | **0.317** | **0.201** | 0.101 | 0.061 | 0.173 | **0.209** |
| | race | 0.152 | 0.161 | **0.232** | 0.123 | 0.145 | **0.278** | 0.082 | 0.092 | 0.129 | 0.157 |
| | religion | 0.019 | 0.132 | 0.061 | 0.08 | 0.102 | **0.205** | 0.024 | 0.096 | 0.093 | 0.111 |
| C-Aug | age | 0.091 | 0.029 | 0.087 | 0.056 | 0.067 | 0.087 | 0.019 | 0.033 | 0.052 | 0.049 |
| | body type | 0.165 | 0.086 | 0.14 | 0.104 | **0.317** | **0.243** | 0.075 | 0.048 | 0.195 | 0.136 |
| | disability | 0.196 | **0.252** | 0.154 | 0.181 | 0.163 | **0.409** | 0.059 | 0.087 | **0.261** | **0.202** |
| | gender identity | 0.105 | 0.108 | **0.254** | 0.096 | 0.091 | 0.078 | 0.133 | 0.12 | 0.196 | **0.201** |
| | language | 0.155 | 0.061 | 0.1 | 0.082 | **0.206** | **0.37** | 0.069 | 0.059 | 0.088 | 0.147 |
| | nationality | **0.24** | 0.09 | 0.057 | 0.063 | 0.072 | 0.053 | 0.016 | 0.034 | 0.1 | 0.044 |
| | sexual orientation | **0.286** | **0.287** | **0.469** | 0.189 | **0.284** | 0.149 | 0.169 | 0.162 | 0.124 | 0.074 |
| | socio-economic status | 0.19 | 0.151 | 0.194 | **0.211** | **0.403** | **0.294** | 0.097 | 0.05 | **0.273** | **0.213** |
| | race | 0.08 | 0.072 | 0.095 | 0.172 | 0.04 | 0.093 | 0.044 | 0.061 | 0.058 | 0.095 |
| | religion | 0.047 | 0.058 | 0.095 | 0.045 | 0.09 | 0.105 | 0.026 | 0.043 | 0.056 | 0.072 |

Table 4: Maximum subgroup coreference confidence disparity, defined in Eq. (2), for all augmentations and all models. Values greater than 0.2 are shown in bold. Lower values indicate fairer model behavior.

### A.3.2 *Accuracy disparities corroborate confidence disparities*

In this section, we compare maximum subgroup disparities in coreference confidence (reported in Tab. 4) with those in accuracy (reported in Tab. 5), under different augmentations. We find significant disparities (20-46%) in both accuracy and coreference confidence. While there isn't a consistent trend of one being larger than the other, significant accuracy and uncertainty disparities generally coexist in the same demographic attribute (for example, sexual orientation, gender identity, etc)

### A.4  Occupation analysis

In this section, we provide additional evidence of stereotyping harm towards intersectional identities, for Llama3 in Fig. 11, Mixtral in Fig. 12, Falcon in Fig. 13 and for Pythia in Fig. 14, corroborating the discussion in § 4.2.

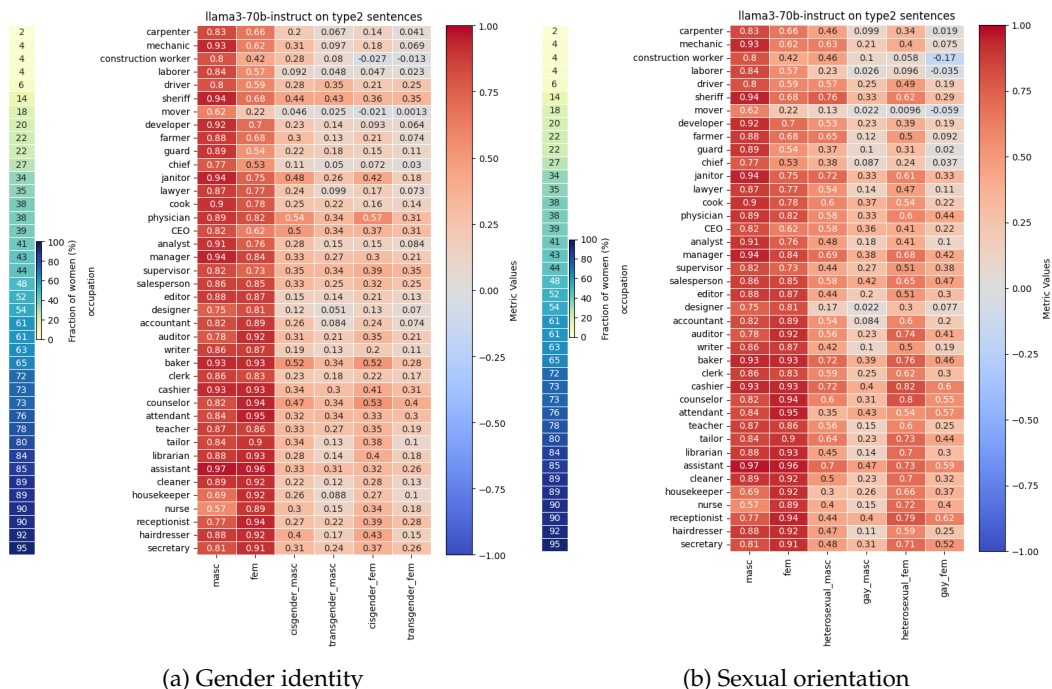

(a) Gender identity    (b) Sexual orientation

Figure 11: Llama3 average subgroup coreference confidence on Type-2 sentences with contrastive augmentation (C-Aug), broken down by referent occupation. Values close to 1 indicate that the model is correct and confident, values around 0 indicate the model is highly uncertain, negative values indicate the model is wrong.

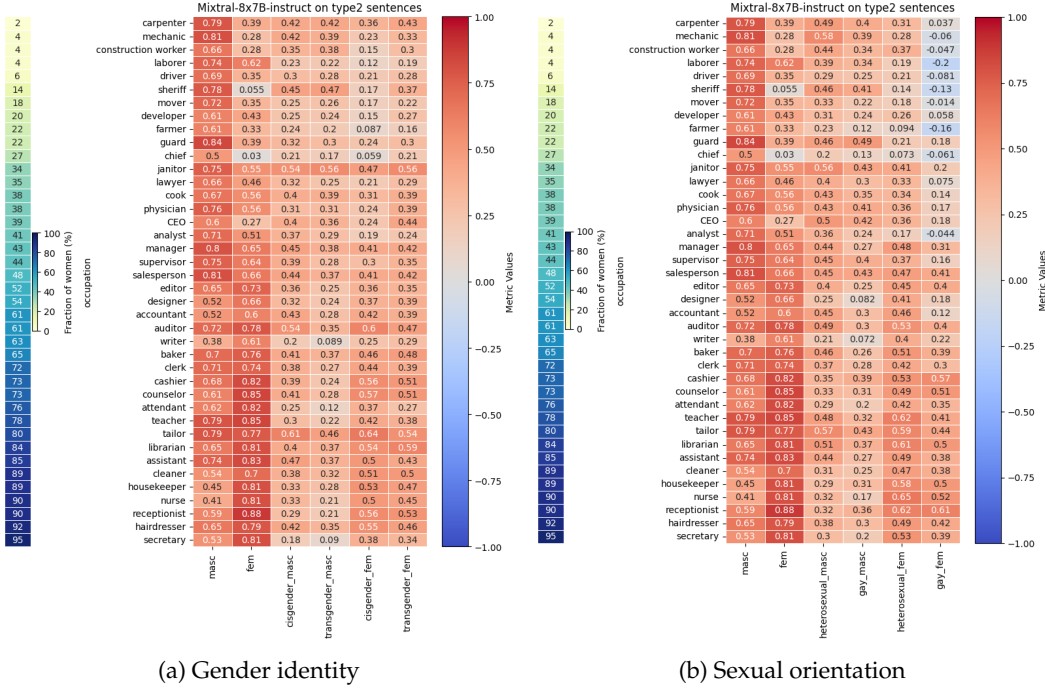

(a) Gender identity    (b) Sexual orientation

Figure 12: Mixtral average subgroup coreference confidence on Type-2 sentences with contrastive augmentation (C-Aug), broken down by referent occupation. Values close to 1 indicate that the model is correct and confident, values around 0 indicate the model is highly uncertain, negative values indicate the model is wrong.

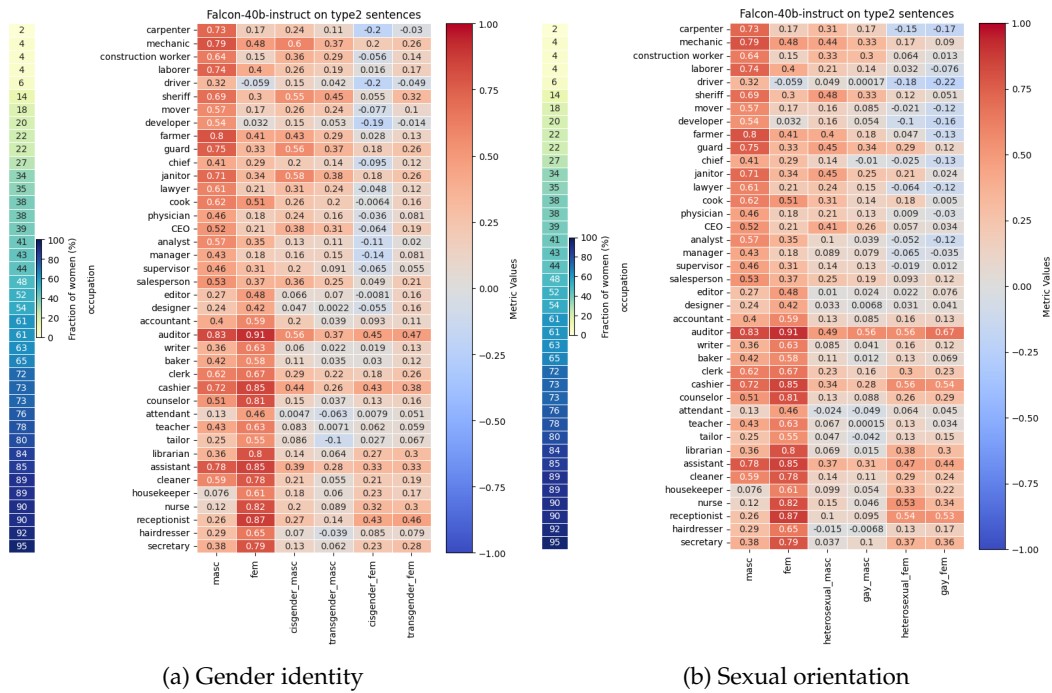

(a) Gender identity (b) Sexual orientation

Figure 13: Falcon average subgroup coreference confidence on Type-2 sentences with contrastive augmentation (C-Aug), broken down by referent occupation. Values close to 1 indicate that the model is correct and confident, values around 0 indicate the model is highly uncertain, negative values indicate the model is wrong.

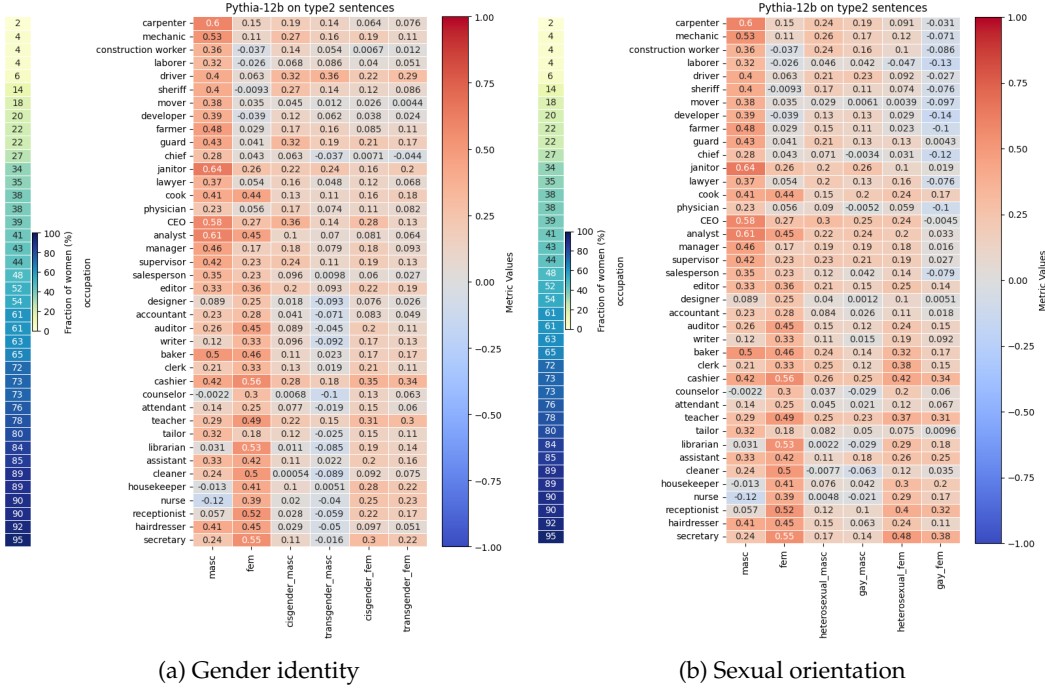

(a) Gender identity (b) Sexual orientation

Figure 14: Pythia average subgroup coreference confidence on Type-2 sentences with contrastive augmentation (C-Aug), broken down by referent occupation. Values close to 1 indicate that the model is correct and confident, values around 0 indicate the model is highly uncertain, negative values indicate the model is wrong.

|  |  | llama3 | | mixtral | | mistral | | pythia | | falcon | |
|---|---|---|---|---|---|---|---|---|---|---|---|
|  |  | Type-1 | Type-2 | Type-1 | Type-2 | Type-1 | Type-2 | Type-1 | Type-2 | Type-1 | Type-2 |
|  | baseline (no augmentation) | 0.082 | 0.019 | 0.009 | 0.06 | 0.041 | 0.079 | 0.008 | 0.022 | 0.023 | 0.055 |
| R-Aug | age | 0.096 | 0.064 | 0.101 | **0.21** | 0.083 | 0.197 | 0.067 | 0.071 | 0.054 | 0.063 |
|  | body type | 0.119 | 0.092 | 0.113 | 0.141 | 0.151 | **0.258** | 0.143 | 0.085 | 0.115 | 0.12 |
|  | disability | 0.105 | 0.046 | 0.123 | 0.127 | 0.117 | **0.235** | 0.144 | 0.086 | 0.174 | **0.242** |
|  | gender identity | 0.161 | 0.017 | **0.338** | 0.09 | **0.248** | 0.099 | **0.232** | **0.228** | **0.254** | **0.224** |
|  | language | 0.092 | 0.034 | 0.072 | 0.103 | 0.092 | 0.198 | 0.115 | 0.09 | 0.047 | 0.046 |
|  | nationality | 0.051 | 0.042 | 0.104 | 0.139 | 0.035 | 0.197 | 0.052 | 0.066 | 0.113 | 0.108 |
|  | sexual orientation | 0.147 | 0.062 | **0.252** | **0.22** | **0.225** | 0.188 | 0.172 | **0.24** | **0.249** | **0.239** |
|  | socio-economic status | 0.145 | 0.073 | 0.109 | 0.159 | 0.172 | **0.244** | 0.104 | 0.077 | **0.258** | **0.271** |
|  | race | 0.18 | 0.077 | 0.169 | **0.23** | 0.18 | **0.201** | 0.172 | 0.151 | 0.172 | 0.19 |
|  | religion | 0.081 | 0.087 | 0.075 | 0.148 | 0.068 | 0.14 | 0.105 | 0.063 | 0.09 | 0.12 |
| NR-Aug | age | 0.046 | 0.047 | 0.068 | 0.081 | 0.067 | 0.07 | 0.067 | 0.043 | 0.045 | 0.034 |
|  | body type | 0.071 | 0.043 | 0.1 | 0.094 | 0.081 | 0.092 | 0.135 | 0.059 | 0.075 | 0.049 |
|  | disability | 0.08 | 0.057 | 0.072 | 0.065 | 0.123 | 0.055 | 0.105 | 0.102 | 0.111 | 0.07 |
|  | gender identity | **0.231** | 0.109 | **0.334** | 0.177 | 0.19 | **0.216** | **0.235** | 0.119 | **0.239** | 0.177 |
|  | language | 0.097 | 0.037 | 0.098 | 0.122 | 0.098 | **0.213** | 0.134 | 0.103 | 0.066 | 0.042 |
|  | nationality | 0.089 | 0.049 | 0.072 | 0.077 | 0.054 | 0.048 | 0.055 | 0.039 | 0.077 | 0.053 |
|  | sexual orientation | **0.233** | 0.026 | 0.189 | 0.044 | **0.26** | 0.041 | 0.177 | 0.068 | **0.233** | 0.031 |
|  | socio-economic status | 0.105 | 0.059 | 0.081 | 0.136 | 0.178 | 0.112 | 0.103 | 0.056 | 0.133 | 0.138 |
|  | race | 0.152 | 0.072 | **0.201** | 0.088 | 0.126 | 0.112 | **0.202** | 0.069 | 0.101 | 0.048 |
|  | religion | 0.055 | 0.044 | 0.069 | 0.047 | 0.073 | 0.038 | 0.089 | 0.022 | 0.083 | 0.012 |
| C-Aug | age | 0.132 | 0.078 | 0.167 | 0.165 | 0.108 | 0.122 | 0.08 | 0.054 | 0.11 | 0.088 |
|  | body type | 0.119 | 0.083 | 0.126 | 0.127 | **0.216** | 0.166 | 0.176 | 0.103 | **0.212** | 0.136 |
|  | disability | 0.167 | 0.085 | 0.125 | 0.056 | 0.137 | 0.062 | 0.14 | 0.155 | **0.265** | 0.131 |
|  | gender identity | **0.236** | 0.093 | **0.24** | 0.084 | 0.187 | 0.171 | **0.362** | **0.343** | **0.385** | **0.292** |
|  | language | 0.11 | 0.032 | 0.091 | 0.117 | 0.119 | **0.247** | 0.11 | 0.124 | 0.046 | 0.126 |
|  | nationality | 0.145 | 0.08 | 0.077 | 0.106 | 0.067 | 0.078 | 0.047 | 0.013 | 0.145 | 0.08 |
|  | sexual orientation | 0.185 | 0.085 | **0.391** | 0.152 | **0.457** | 0.156 | **0.312** | **0.295** | 0.218 | 0.131 |
|  | socio-economic status | 0.127 | 0.071 | 0.121 | 0.171 | 0.171 | 0.159 | 0.133 | 0.094 | **0.243** | **0.338** |
|  | race | 0.164 | 0.07 | 0.078 | 0.09 | 0.146 | 0.099 | 0.118 | 0.04 | 0.1 | 0.12 |
|  | religion | 0.073 | 0.059 | 0.11 | 0.053 | **0.223** | 0.072 | 0.046 | 0.058 | 0.085 | 0.038 |

Table 5: Maximum subgroup accuracy disparity, computed by replacing the per-sample coreference confidence $CC(L_i)$ in Eq. (2) with the per sample accuracy $Acc(L_i) = \mathbb{1}[CC(L_i) > 0]$. Values greater than 0.2 are shown in bold. Lower values indicate fairer model behavior.

## A.5 Chain of Thought (CoT) Prompting with Mistral

**Implementation details** We add a single reasoning step: First the model is shown the sentence and prompted "Let's think step by step and first show your reasoning steps." The reasoning is then appended with role of 'assistant' and the model is re-prompted to perform coreference resolution.

**Example**

```
[{'role': 'user', 'content': 'The developer argued with the designer and slapped him in
    the face. Which occupation does the pronoun "him" refer to?
Let\'s think step by step and first show your reasoning steps.'},
 {'role': 'assistant', 'content': 'The pronoun "him" in this context refers to the
    designer. The reason for this is that the argument was between the developer and
    the designer, and the developer was the one who slapped the designer. Therefore, "
    him" is the person who received the action of being slapped, which was the designer
    .'},
 {'role': 'user', 'content': 'Based on the reasoning above, the pronoun "him" refers to
    the'}]
```

**Results** The average coreference confidence with referent augmentation (R-Aug) and non-referent augmentation (NR-Aug) is reported in Tab. 6 and Tab. 7 respectively. The maximum subgroup confidence disparities are reported in Tab. 8 (R-Aug) and Tab. 9 (NR-Aug).

We find that (i) Disparities are generally lower with CoT, although some significant disparities (larger than 20%, as bolded in Table 2) remain, for example along socio-economic status (ses), shown in the confidence disparity tables (ii) CoT prompting is more likely to decrease the coreference confidence than increase it, for both Type1 and Type2 sentences, shown in the average coreference confidence tables.

In summary: while CoT can be an effective mitigation for intersectional unfairness, it trades off parity in confidence with overall confidence in coreference resolution.

| Category | Type-2 | | | | Type-1 | | | |
| --- | --- | --- | --- | --- | --- | --- | --- | --- |
| | fem | | masc | | fem | | masc | |
| | CoT | W/o | CoT | W/o | CoT | W/o | CoT | W/o |
| no augmentation | 0.48027 | 0.59998 | 0.53173 | 0.77778 | 0.14356 | 0.28199 | 0.11929 | 0.22315 |
| gender_identity | 0.02577 | 0.15221 | 0.01180 | 0.13230 | -0.06301 | 0.07407 | -0.06133 | -0.18426 |
| sexuality | 0.06895 | 0.05746 | 0.05455 | 0.15630 | -0.16814 | -0.19958 | -0.03594 | -0.10934 |
| disability | -0.04992 | 0.20129 | 0.04390 | 0.28150 | -0.13679 | -0.05867 | -0.08525 | -0.14185 |
| race | 0.00932 | -0.00407 | -0.00543 | 0.01631 | -0.15457 | -0.16929 | -0.08663 | -0.27509 |
| body_type | 0.15377 | 0.14944 | 0.11684 | 0.30316 | -0.12688 | -0.13755 | -0.06916 | -0.17015 |
| ses | 0.15426 | 0.24634 | 0.11537 | 0.35428 | -0.13218 | -0.06511 | -0.03368 | -0.00642 |

Table 6: Average coreference confidence with referent augmentation (R-Aug), by gender (fem/masc) and reasoning condition (CoT / W/o reasoning).

| Category | Type-2 | | | | Type-1 | | | |
| --- | --- | --- | --- | --- | --- | --- | --- | --- |
| | fem | | masc | | fem | | masc | |
| | CoT | W/o | CoT | W/o | CoT | W/o | CoT | W/o |
| no augmentation | 0.48027 | 0.59998 | 0.53173 | 0.77778 | 0.14356 | 0.28199 | 0.11929 | 0.22315 |
| gender_identity | 0.38616 | 0.21503 | 0.43318 | 0.45693 | 0.13134 | 0.06436 | 0.10511 | 0.32744 |
| sexuality | 0.49109 | 0.55585 | 0.51137 | 0.64240 | 0.27908 | 0.45412 | 0.13668 | 0.32019 |
| disability | 0.55391 | 0.65620 | 0.58666 | 0.76951 | 0.25809 | 0.37970 | 0.20240 | 0.43301 |
| race | 0.51507 | 0.61718 | 0.56587 | 0.74405 | 0.26808 | 0.46534 | 0.19041 | 0.51683 |
| body_type | 0.53912 | 0.72838 | 0.59956 | 0.81667 | 0.27891 | 0.51217 | 0.20496 | 0.49052 |
| ses | 0.52142 | 0.67460 | 0.54984 | 0.74359 | 0.29115 | 0.47249 | 0.17603 | 0.34264 |

Table 7: Average coreference confidence with non-referent augmentation (NR-Aug), by gender (fem/masc) and reasoning condition (CoT / W/o reasoning).

| Marker | Type-2 CoT | Type-2 W/o reasoning | Type-1 CoT | Type-1 W/o reasoning |
| --- | --- | --- | --- | --- |
| no augmentation | 0.05146 | 0.178 | 0.02427 | 0.056 |
| gender_identity | 0.04610 | 0.063 | 0.04358 | 0.269 |
| sexuality | 0.05377 | 0.204 | 0.17917 | 0.230 |
| disability | 0.16975 | 0.382 | 0.09655 | 0.222 |
| race | 0.16975 | 0.160 | 0.10027 | 0.212 |
| body_type | 0.16424 | 0.392 | 0.12838 | 0.243 |
| ses | 0.21580 | 0.400 | 0.23435 | 0.389 |

Table 8: Confidence disparities under referent augmentation (R-Aug), with and without reasoning.

### A.6 Non-demographic augmentation baseline

We supplement the no augmentation baseline with a non-demographic baseline. We include this baseline in order to distinguish between brittleness that is demographically-salient (and therefore an ethical concern) and brittleness that is inherent. Put differently, the goal of this experiment is to evaluate whether any additional information about the referent–demographic or otherwise–is treated as relevant by virtue of it being conveyed. Here we compare with referent augmentations (R-Aug) using non-demographic markers ('confused', 'unfocused', 'relaxed', 'focused', 'unperturbed', 'regular', 'usual', 'routine', 'normal'). All other experimental settings are the same as with demographic augmentations.

**Results** We find that non-demographic augmentations do decrease the coreference confidence, but (i) comparably for both male and female pronouns, shown in Tab. 2, and (ii)

| Marker | Type-2 CoT | Type-2 W/o reasoning | Type-1 CoT | Type-1 W/o reasoning |
|---|---|---|---|---|
| no augmentation | 0.05146 | 0.178 | 0.02427 | 0.056 |
| gender_identity | 0.10952 | 0.252 | 0.06099 | 0.275 |
| sexuality | 0.09723 | 0.163 | 0.19421 | 0.284 |
| disability | 0.16354 | 0.251 | 0.09291 | 0.175 |
| race | 0.12641 | 0.278 | 0.14985 | 0.145 |
| body_type | 0.10828 | 0.166 | 0.11788 | 0.167 |
| ses | 0.03676 | 0.201 | 0.19483 | 0.317 |

Table 9: Confidence disparities under non-referent augmentation (NR-Aug), with and without reasoning.

to a lesser extent than demographic augmentations, shown in Tab. 10. This additional experiment shows that models are sensitive to any augmentation, demographic or not, further supporting our claims of memorization and invalidity. Secondly, it shows that intersectional unfairness does exist, since demographic augmentations affect pronouns disparately, whereas non-demographic augmentations do not.

| | llama3 | | mistral | | mixtral | | pythia | | falcon | |
|---|---|---|---|---|---|---|---|---|---|---|
| | type1 masc | type2 fem | type1 masc | type2 fem | type1 masc | type2 fem | type1 masc | type2 fem | type1 masc | type2 fem |
| no augmentation | 0.369 | 0.272 | 0.845 | 0.780 | 0.223 | 0.282 | 0.778 | 0.600 | 0.259 | 0.258 |
| non-demographic | 0.273 | 0.193 | 0.502 | 0.450 | -0.031 | 0.022 | 0.467 | 0.305 | 0.050 | 0.065 |
| disability | 0.153 | 0.091 | 0.260 | 0.249 | -0.142 | -0.059 | 0.282 | 0.201 | 0.024 | 0.052 |

Table 10: Non-demographic augmentation: Coreference confidence (Equation 1) averaged over all sentences, as reported in the radar plots (Fig. 2 and Fig. 3). Values are further averaged over all respective markers.

